# Generation and Characterization of an Influenza D Reporter Virus

**DOI:** 10.3390/v15122444

**Published:** 2023-12-16

**Authors:** Lukas Probst, Laura Laloli, Manon Flore Licheri, Matthias Licheri, Mitra Gultom, Melle Holwerda, Philip V’kovski, Ronald Dijkman

**Affiliations:** 1Institute for Infectious Diseases, University of Bern, 3001 Bern, Switzerland; 2Graduate School for Cellular and Biomedical Sciences, University of Bern, 3012 Bern, Switzerland; 3Multidisciplinary Center for Infectious Diseases, University of Bern, 3012 Bern, Switzerland; 4European Virus Bioinformatics Center, 07743 Jena, Germany; 5Microscope Imaging Center, University of Bern, 3012 Bern, Switzerland

**Keywords:** Influenza D Virus (IDV), reverse genetics, fluorescent reporter virus, antivirals

## Abstract

Influenza D virus (IDV) can infect various livestock animals, such as cattle, swine, and small ruminants, and was shown to have zoonotic potential. Therefore, it is important to identify viral factors involved in the broad host tropism and identify potential antiviral compounds that can inhibit IDV infection. Recombinant reporter viruses provide powerful tools for studying viral infections and antiviral drug discovery. Here we present the generation of a fluorescent reporter IDV using our previously established reverse genetic system for IDV. The mNeonGreen (mNG) fluorescent reporter gene was incorporated into the IDV non-structural gene segment as a fusion protein with the viral NS1 or NS2 proteins, or as a separate protein flanked by two autoproteolytic cleavage sites. We demonstrate that only recombinant reporter viruses expressing mNG as an additional separate protein or as an N-terminal fusion protein with NS1 could be rescued, albeit attenuated, compared to the parental reverse genetic clone. Serial passaging experiments demonstrated that the mNG gene is stably integrated for up to three passages, after which internal deletions accumulate. We conducted a proof-of-principle antiviral screening with the established fluorescent reporter viruses and identified two compounds influencing IDV infection. These results demonstrate that the newly established recombinant IDV reporter virus can be applied for antiviral drug discovery and monitoring viral replication, adding a new molecular tool for investigating IDV.

## 1. Introduction

Influenza D virus (IDV) belongs to the family *Orthomyxoviridae* and the genus Deltainfluenzavirus and was discovered in 2011 in an infected pig with flu-like symptoms [1]. Cattle were later identified as the main host reservoirs of IDV; however, several seroepidemiological studies have shown indirect evidence of IDV infections in a large variety of species, including small ruminants, camels, and humans with close occupational contact with cattle [2,3,4,5,6,7]. This broad host tropism, the ability of IDV to infect primary human airway epithelial cell (hAEC) cultures, and the detection of IDV in nasal washes of cattle farmers raise concern about the zoonotic potential of IDV and warrant further studies to better understand the interaction between IDV and its different hosts [8,9].

Reverse genetics systems are valuable tools for studying viruses and their interactions with host determinants. They allow for targeted mutagenesis of the viral proteins for functional studies and the incorporation of reporter genes encoding fluorescent or bioluminescent proteins into the viral genome. In particular, fluorescent reporter viruses have been widely used as powerful tools for studying in vitro and in vivo dynamics of Influenza A virus (IAV) infections and evaluating treatment options [10,11,12,13,14]. Different strategies for introducing reporter genes into the segmented IAV genome have been established, including introducing eGFP into the shortest segment encoding the non-structural proteins NS1 and NS2 [10,11,12,13,14,15]. During the virus replication cycle, the NS1 protein is translated directly from an unspliced mRNA transcript and functions as a suppressor of the innate immune system. In contrast, the NS2 protein is generated through alternative splicing and is hypothesized to facilitate the nuclear export of newly produced viral ribonucleoproteins (vRNPs) during influenza virus replication [16,17]. Similar reporter viruses have been generated for the human pathogen Influenza B virus (IBV) [18,19]. Recently, we and others established reverse genetics systems for IDV, facilitating targeted molecular studies of this virus; however, to date, no reporter virus has been described for IDV [20,21,22,23].

This study aimed to generate a fluorescent reporter IDV utilizing our established reverse genetics system to facilitate further studies of IDV, such as inhibitor screenings. To achieve this, we introduced the fluorescent reporter mNeonGreen (mNG) [24] into the shortest segment of IDV, the NS segment. We rescued two out of three different reporter viruses and demonstrated that the fluorescent gene was stably integrated for the first three passages after which deletions in the reporter gene accumulated. We used the newly generated reporter virus to rapidly screen a panel of compounds for antiviral properties and demonstrate the applicability of the reporter virus in large-scale antiviral compound screenings.

## 2. Materials and Methods

### 2.1. Cells

Human rectal tumor 18G cells (HRT-18G, American Type Culture Collection (ATCC), Manassas, VA, USA) were cultured in Dulbecco’s minimum essential medium containing Glutamax (DMEM + Glutamax; Gibco, Gaithersburg, MD, USA) and supplemented with 5% heat-inactivated Fetal Bovine Serum (FBS, PAN Biotech, Aidenbach, Germany), 100 μg/mL streptomycin, and 100 IU/mL penicillin (Gibco). The human embryonal kidney (HEK) cell line 293-LTV (LTV-100; Cellbiolabs, San Diego, CA, USA) was maintained in Dulbecco’s minimum essential medium containing Glutamax (DMEM + Glutamax; Gibco) and was additionally supplemented with 10% FBS (PAN Biotech), 1% minimum essential medium non-essential amino acids (MEM NEAA (100×), Gibco), 100 μg/mL streptomycin, and 100 IU/mL penicillin (Gibco). Both cell lines were cultured at 37 °C in a humidified incubator with 5% CO_2_.

### 2.2. Construction of the NS Reporter Plasmid

The coding sequences for the reporter segments (NS1-2A-mNG-2A-NS2, NS1-mNG-2A-NS2, NS1-2A-mNG-NS2), including the non-coding end-regions were custom synthesized and cloned into a pcDNA3.1 backbone (Genscript, Piscataway, NJ, USA). The individual NS reporter segments were cloned into the bidirectional pHW2000 backbone using the in vivo assembly (IVA) method [25]. The NS reporter segment was amplified by PCR using ExTaq polymerase (Clontech, Mountain View, CA, USA) following the manufacturer’s protocol using the primers NS1_IDV_Seg_PCR Fw and NS1_IDV_Seg_PCR Rv at 100 nM and the following cycle profile: 98 °C for 1 min followed by 30 cycles of 98 °C for 10 s, 55 °C for 30 s, 72 °C for 2 min and 30 s, and a final elongation step at 72 °C for 5 min. After PCR amplification, the amplicons were treated with 20 U DpnI (New England Biolabs, Ipswich, MA, USA) for 1 h at 37 °C to remove the template plasmid and purified using the E.Z.N.A. Cycle pure Kit (Omega Bio-Tek, Norcross, GA, USA) according to the manufacturer’s guidelines.

The backbone of the pHW2000 plasmid (kindly provided by Prof. Dr. Martin Schwemmle, University of Freiburg, Germany) was amplified using the ExTaq polymerase (Clontech) following the manufacturer’s protocol using the primers pHW2000_LF_IVD1_a and pHW2000_RF_IVD_s at 100 nM and a 2-step protocol: 98 °C for 2 min followed by 30 cycles of 98 °C for 10 s and 68 °C for 4 min. The PCR amplicon was treated with DpnI as described above and purified on a 1% agarose gel using the E.Z.N.A. Gel extraction kit (Omega Bio-Tek), according to the manufacturer’s protocol.

The PCR products were mixed in an insert:vector ratio of 5:1 and transformed to Stellar competent cells (Clontech). Colonies carrying the desired plasmids were identified by colony PCR using the GoTaq green master mix (Promega, Madison, WI, USA) in accordance with the manufacturer’s protocol using the primers CMV-for and BGH-rev at 400 nM and the following cycle profile: 94 °C for 5 min followed by 25 cycles of 94 °C for 1 min, 55 °C for 1 min, 72 °C for 2 min, and a final elongation of 7 min at 72 °C. Positive colonies were grown overnight in an LB medium containing 50 µg/mL Ampicillin, and the plasmids were extracted with the E.Z.N.A. plasmid Midi Kit (Omega Bio-tek) according to the manufacturer’s protocol. All plasmids were verified by Sanger sequencing (Microsynth, Balgach, Switzerland). The primer sequences are listed in Appendix A.

### 2.3. Western Blot

To assess the cleavage efficiency at the 2A sites, we overexpressed the reporter constructs and detected the mNG, NS1-mNG, and mNG-NS2 fusion proteins on a Western blot. The reporter plasmids were mixed at a ratio of 2 µL Lipofectamine 3000 (Invitrogen, Waltham, MA, USA) per 1 µg plasmid and incubated for 30 min at room temperature. The transfection mix was added to 293-LTV cells, which were seeded in a 6-well plate the day before at a density of 3 × 10^5^ cells/well, and incubated at 37 °C. After three days, the cells were lysed using M-PER Mammalian Protein Extraction Reagent (Thermo Fisher Scientific, Waltham, MA, USA) supplied with 1× cOmplete—Mini protease inhibitor cocktail (Roche, Basel, Switzerland). The lysate was mixed with a final concentration of 1× Lämmli-buffer (5 mM Tris pH 6.8, 0.8% glycerol, 0.25% SDS, 0.5% β-mercaptoethanol, and 0.005% bromophenol blue), boiled for 5 min at 95 °C, and quenched on ice. The samples were loaded onto a 4–12% gradient SurePAGE, Bis-Tris precast gel (Genscript) and run in Tris-MOPS-SDS Running Buffer (Genscript) at 180 V for 45 min. Transfer of the proteins onto a nitrocellulose membrane was performed with the eBlot L1 Fast Protein Transfer System according to the manufacturer’s protocol (Genscript). Following the transfer, the membrane was blocked using 5% BSA (IgG-free, Protease-free, Jackson ImmunoResearch, Westgrove, PA, USA) in TBST (20 mM Tris, 500 mM NaCl, pH7.4, 0.05% Tween-20) and then incubated overnight at 4 °C with the primary antibody mouse-anti-mNeonGreen (Chromotek, Planegg, Germany, 1:1000 in TBST with 0.5% BSA). For detection, the blot was incubated with the secondary antibody donkey anti-mouse HRP-conjugated (Jackson ImmunoResearch, 1:5000 in TBST with 0.5% BSA) at room temperature for 1 h. Visualization was achieved using the WesternBright Quantum detection kit (Advansta, San Jose, CA, USA) and imaged using the Fusion FX (Vilber, Collégien, France). For loading control, the membrane was stripped with stripping solution (0.2 M Glycine, 0.1% SDS, 1% Tween20, pH 2.2) twice for 10 min, followed by blocking and incubation with the primary conjugated antibody anti-β-actin HRP-conjugated (Sigma-Aldrich, St. Louis, MO, USA, 1:20,000 in TBST with 0.5% BSA).

### 2.4. Viral Rescue

One day before the transfection, a co-culture of HRT-18G and 293-LTV cells was seeded at a density of 1 × 10^6^ cells of each cell type per well in a 6-well plate. The pHW-plasmids coding for the PB2, PB1, P3, NP, HEF, and P42 segments of the Swiss isolate D/CN286 from our previous study were mixed with one of the plasmids coding for the reporter NS segment at equal amounts of 1 µg each [20]. Lipofectamine 3000 (Invitrogen) was added at a ratio of 2 µL/1 µg DNA and incubated for 30 min at room temperature. The transfection mix was added dropwise to the co-culture and incubated at 37 °C. Two days post-transfection, the cells were washed twice with PBS and supplemented with infection medium (iMEM, Minimum essential medium (MEM + GlutaMax, Gibco) with 5 g/L BSA (Sigma-Aldrich), 15 mM of HEPES (Gibco), 100μg/mL of penicillin and 100 IU of streptomycin (Gibco), and 0.25 µg/mL of bovine-pancreas-isolated trypsin (Sigma-Aldrich). The cells were incubated for 5 days at 37 °C, 5% CO_2_, and on day 3, 0.25 μg/mL fresh trypsin was added. On day 5, the virus-containing supernatant was collected and centrifuged at 1000× rcf for 5 min to remove cells and debris. To propagate the rescued virus, the collected supernatant was diluted 1:4 in iMEM and used to inoculate new HRT-18G cells, which were seeded the day before in a 6-well plate at a density of 1 × 10^6^ cells per well. The cells were incubated at 37 °C and 5% CO_2_ for 7 days. On days 3 and 5, 0.25 μg/mL fresh bovine-pancreas-isolated trypsin was added. After 7 days, the supernatant was collected and titrated using the tissue culture infectious dose 50 (TCID_50_) method, as described below, using the mNG fluorescent signal as readout.

### 2.5. Viral Titration

HRT-18G cells were seeded at a density of 4 × 10^4^ cells per well in 96-well plates the day before infection. The following day, the cells were washed twice with PBS and 50 µL of iMEM was added. The virus-containing supernatant samples were diluted in a 10-fold serial dilution in iMEM, and 50 µL was added to the prepared cells in 96-well plates with 6 replicates. The cells were incubated at 37 °C for 6 days before readout. As readout, the infected cells were detected via mNG fluorescent signal or fixed and stained for IDV NP protein as described below. The viral titer was calculated according to the protocol by Spearman–Kärber [26].

### 2.6. Immunofluorescence

The samples were fixed using 4% formalin for 20 min, washed 3× with PBS, and blocked overnight in CB buffer (PBS supplemented with 50 mM NH_4_Cl, 0.1% Saponin, and 2% BSA (IgG-free, Protease-free)). The cells were stained for the NP protein of IDV using rabbit anti-IDV NP (Genscript, 1:1000 in CB buffer) or mNeonGreen using mouse anti-mNG (Chromotek, 1:500 in CB buffer) for 2 h at room temperature, followed by secondary antibodies donkey anti-Rabbit Alexa Fluor 594 (Jackson ImmunoResearch, 1:400 in CB buffer), donkey anti-Mouse Alexa Fluor 647 (Jackson ImmunoResearch, 1:400 in CB buffer), or donkey anti-Rabbit Alexa Fluor 647 (Jackson ImmunoResearch, 1:2000 in CB buffer) for 1h. The samples were counterstained using 4′,6-diamidino-2-phenylindole (DAPI, Thermo Fisher Scientific, Waltham, MA, USA) and visualized using a Cytation 5 Cell Imaging Multimode Reader (Agilent BioTek, Sursee, Switzerland) equipped with 4× (numerical aperture (NA): 0.13), 10× (NA: 0.3), and 40× (NA: 0.6) air objectives. Images were further processed with Fiji (v1.53q); the brightness and contrast of all images were adjusted identically to the corresponding controls [27]. Figures were generated using the FigureJ plugin (v1.36) [28].

### 2.7. Viral Replication Kinetics

HRT-18G cells were seeded the day before infection at a density of 1.7 × 10^5^ cells per well in a 24-well plate. The cells were infected at an MOI of 0.01 for 1.5 h with IDV^mNG^, IDV^NS1-mNG^, and IDV^WT^. The supernatant was removed and the cells were washed 3× with PBS and supplemented with virus infection medium. The last wash, as well as 100 µL of the supernatant collected every 24 h up to 120 hpi, was stored at −80 °C. At every time point, the removed supernatant was replaced with fresh infection medium. The collected samples were titrated using the TCID_50_ method described above and stained for IDV NP for the readout. In addition, at every time point, pictures were taken with a Cytation 5 Cell Imaging Multimode Reader to monitor the development of the mNG fluorescent signal. The number of mNG-positive cells was quantified using the Gen5 Image Prime software package (v3.11).

### 2.8. Virus Passaging

HRT-18G cells were seeded the day before infection at a density of 1.7 × 10^5^ cells per well in a 24-well plate. The cells were infected at either MOI of 0.1 or 0.01 and incubated at 37 °C. After 6 days, the supernatant was removed and titrated using the TCID_50_ method, with the mNG fluorescence as readout. The supernatant was used to infect new HRT-18G cell cultures for another round of infection at MOI 0.1 and 0.01, for a total of 5 passages. Viral RNA was extracted from all the samples using the E.Z.N.A. viral RNA extraction kit (Omega Bio-tek), with DNAse treatment using the E.Z.N.A. RNAse free DNAse I kit (Omega Bio-tek) according to the manufacturer’s protocol. cDNA synthesis was performed using Lunascript RT SuperMix Kit (New England Biolabs), followed by PCR using the Q5 Hot Start High-Fidelity DNA Polymerase (New England Biolabs) and specific primers targeting the NS segment according to the following protocol: 30 s at 98 °C followed by 35 cycles of 98 °C for 10 s, and 72 °C for 1 min 40 s and a final extension of 2 min at 72 °C. The PCR product was purified using the E.Z.N.A. PCR cycle pure kit (Omega Bio-tek) according to the manufacturer’s protocol. The PCR product (100 ng per sample) was barcoded using the Ligation Sequencing Kit (SQK-LSK109, Oxford Nanopore Technologies (ONT), Oxford, United Kingdom) in combination with native barcodes (EXP-NBD104 and EXP-NBD114, ONT) according to the manufacturer’s instructions. The samples were sequenced on a MinION device (Mk1B, ONT) using a Flongle flow cell (R9.4.1, FLO-FLG001, ONT) with real-time high-accuracy base-calling (Guppy 5.0.11, GPU enabled) enabled in the MinKNOW core (4.3.4). After filtering out reads smaller than 200 bp, 100 reads were randomly subsampled using seqkit (v2.1.0) [29]. The reads were mapped using minimap2 (splicing enabled, v2.21-r1071), and sequencing depth was calculated using samtools (v1.10). Figures were generated using R (version 4.3.1).

### 2.9. Plaque Purification

HRT-18G cells were seeded the day before infection at a density of 1.7 × 10^5^ cells per well in a 24-well plate. A 10-fold serial dilution of the virus was prepared and used to infect the prepared HRT-18G cells. After 1 h, the supernatant was removed and the cells were overlaid with 0.6% (*w*/*v*) low melting-point agarose (Sigma-Aldrich) diluted in iMEM. The agarose was solidified at room temperature, and cells were subsequently incubated at 37 °C for 3 days. The mNG-positive plaques were picked under a fluorescence microscope and transferred into 200 µL iMEM. A 150 µL volume of each picked plaque was used to infect new HRT-18G cells, which were seeded the day before in a 6-well plate at a density of 1 × 10^6^ cells per well. After 5 days, the supernatant was collected and titrated using the tissue culture infectious dose 50 (TCID_50_) method, with the mNG fluorescent signal as readout.

### 2.10. Antiviral Compound Screening

HRT-18G cells were seeded at a density of 4 × 10^4^ cells per well in black-wall, clear-bottom 96-well plates (Costar, Corning, NY, USA) the day before the experiment. The cells were washed twice with PBS and supplemented with an infection medium containing the tested compounds and incubated at 37 °C. After 1 h, IDV^mNG^ of passage 2 was added at a MOI of 0.01 and the cells were incubated further at 37 °C for 3 days. At 3 days post-infection, images were acquired using Cytation 5 Cell Imaging Multimode Reader, and the number of mNG positive cells was counted using the Gen5 ImagePrime software package (v3.11). Cell viability was determined using the CellTiter-Glo Luminescent Cell Viability Assay (Promega) according to the manufacturer’s guidelines. Z-scores were determined for cell viability and number of infected cells using R (version 4.0.2). Compounds with a cell viability z-score >2 or <−2 were considered toxic and excluded from downstream analysis. Compounds with a z-score >1.5 or <−1.5 in the number of mNG-positive cells were considered as hits and further tested in a 2-fold serial dilution spanning concentrations from 2 to 0.008 µM and infected with IDV^mNG^ at an MOI of 0.01 to determine dose–response curves. The dose–response curves and EC_50_ values of the hit compounds were determined in R (version 4.3.1) using the drc package [30]. The compounds tested were Baloxavir Marboxil (MedChemExpress, Monmouth Junction, NJ, USA) and a panel of 80 available compounds from the COVID box (Medicines for Malaria Venture, Geneva, Switzerland) at a concentration of 1 µM (Appendix A) [31]. To confirm the effects on IDV^WT^ replication, the dose–response experiments were repeated with IDV^WT^ infection at MOI of 0.01, and the cells were fixated and NP immunostained to detect infected cells, as described above.

## 3. Results

### 3.1. Generation of a Reporter IDV

To generate an IDV reporter virus, we introduced the mNeonGreen (mNG) fluorescent reporter gene into the shortest segment of the virus, NS, between nonstructural protein 1 (NS1) and NS2, from which the latter is generated via alternative splicing. To generate a non-spliced mRNA transcript encoding both viral proteins and the mNG reporter protein, we disrupted the splice sites required to generate the NS2 transcript and duplicated the entire NS2 coding sequence (Figure 1A). To separate the individual proteins during translation, we flanked the mNG fluorescent protein with PTV-2A autoproteolytic cleavage sites [32]. In addition, we generated reporter constructs in which one of the 2A cleavage motifs was removed to produce either NS1-mNG or mNG-NS2 fusion proteins. The synthetic DNA constructs were subcloned into the corresponding pHW2000 backbone to generate recombinant viruses using our previously established reverse genetics system for IDV [20].

First, we confirmed the autoproteolytic cleavage functionality and efficiency of the transfected constructs using Western blot (Figure 1B). Using an antibody directed at the mNG protein, we confirmed that the reporter construct with the two PTV-2A sites flanking the mNG protein (IDV^mNG^) yielded a single protein product 27 kDa in size corresponding to mNG. For the NS1-mNG (IDV^NS1-mNG^) and the mNG-NS2 (IDV^mNG-NS2^) reporter fusion constructs, a single band was visible at 60 or 52 kDa, respectively, as expected. However, the signal intensity for the band corresponding to the mNG-NS2 fusion protein was much lower than the signal intensities detected for the other two reporter constructs (Figure 1B), suggesting that only two out of the three constructs might yield viable replicating reporter viruses.

Next, the viral rescue of the three different constructs was performed by transfecting the corresponding IDV rescue plasmids into a co-culture of HRT-18G/293-LTV cells [20]. The supernatant was transferred 6 days post-transfection on to fresh HRT-18G cells, after which mNG expression from the different reporter viruses was assessed using fluorescence microscopy 6 days post-infection (dpi). This revealed that mNG-positive cells could be detected for IDV^mNG^ and IDV^NS1-mNG^, but not for IDV^mNG-NS2^. This is in line with the results obtained in the previous transfection experiment and indicates that efficient protein expression and cleavage are required to rescue the IDV reporter virus. Using a nucleoprotein (NP)-specific antibody, we confirmed that mNG expression of both IDV^mNG^ and IDV^NS1-mNG^ coincided with NP expression, indicating that the fluorescent reporter is expressed in all IDV ^NG^ and IDV^NS1-mNG^-infected HRT-18G cells (Figure 1C).

Subsequently, we compared the viral replication kinetics of the two rescued reporter viruses with the parental IDV strain (IDV^WT^). We inoculated HRT-18G cells with MOI 0.01 and collected supernatant samples every 24 h for 120 h to quantify viral replication via virus titration. This revealed similar replication kinetic profiles for all viruses, characterized by a strong increase in viral titers during the first 48 h, after which a plateau is reached at 72 hpi. It is important to note that the viral titers of the two reporter viruses were 1 to 2 orders of magnitude lower than the parental IDV^WT^ (Figure 1D) at all time points. In addition to the viral titers, we monitored the fluorescent signal development over time to quantify the total number of mNG-positive cells for the IDV^mNG^ and IDV^NS1-mNG^ reporter viruses with 24 h intervals (Figure 1D). This revealed that the first mNG-positive cells could be detected at 24 hpi, increasing rapidly from 48 hpi and reaching a plateau at 96 hpi. These results show that the fluorescent mNG signal follows the same dynamics as the viral titer and can be used to detect and quantify infections with the reporter IDV.

### 3.2. mNG Gene Expression Is Maintained for the First Three Passages

To study the stability of the integrated reporter gene, we passaged the recombinant reporter viruses at two different MOIs (0.1 and 0.01) on HRT-18G cells and titrated the supernatant containing the different passages after 6 dpi using the mNG fluorescent signal as a readout. We observed similar titers for the first three passages, followed by a rapid decline in passages 4 and 5, independent of the MOI used (Figure 2A). To determine whether the observed loss of titer based on mNG fluorescence is caused by decreased infectivity of the reporter virus or by the loss of mNG expression, we immunostained reporter-virus-infected HRT-18G cells at different passages for viral NP and mNG protein expression (Figure 2B and Appendix A). This demonstrated that the NP and mNG fluorescent signals overlap in early passages, but decrease in later passages, indicating a potential loss of the mNG coding sequence in the reporter virus. To confirm this, we extracted the viral RNA from the different passages and sequenced the amplified NS segment in full using Oxford Nanopore sequencing technology (Figure 2C and Appendix A). This approach revealed that the full-length segment was present during the first passages, but large internal deletions accumulated during later passages. In this analysis, no defined genetic breaking points were identified when comparing the different passaging experiments, suggesting random deletion events rather than an accidental introduction of a novel splice site or defined recombination. These results also demonstrated that the fluorescent readout can be used with recombinant reporter viruses passaged no more than three times.

### 3.3. Plaque Purification Increases Retention of the Full-Length Reporter Segment

To select reporter viruses with an intact mNG reporter gene, we performed two rounds of plaque purification and subsequent expansion starting either directly with the rescued virus (P0) or the passage 2 stocks (P2). Sequencing of the resulting viruses showed that the plaque-purified viruses maintained an intact reporter segment up to the last passage (Figure 3A and Appendix A). Additionally, starting with a P2 stock that already contained internal deletions, we demonstrated that reporter viruses with an intact mNG reporter can be isolated via plaque purification from a heterologous population already carrying internal deletions. This was corroborated by immunostaining cells infected with plaque-purified reporter viruses for mNG and NP expression (Figure 3B and Appendix A). These results demonstrate that plaque purification of the rescued reporter virus is essential to selecting viruses with an intact reporter gene.

### 3.4. Identification of Compounds Inhibiting and Promoting Viral Replication

Because recombinant fluorescent reporter viruses are powerful tools for identifying novel antiviral compounds, we performed a proof-of-concept study using our IDV^mNG^ fluorescent reporter virus from passage 2 to test a panel of 80 compounds from the COVID box (Appendix A). Here we included Baloxavir Marboxil as a positive control, as it was previously shown to inhibit IDV replication [31]. Using a similar strategy, which we previously employed for SARS-CoV-2, we first evaluated cell viability after an incubation period of 73 h using the 80 compounds and the Baloxavir Marboxil at concentrations of 1 µM and the vehicle control (DMSO) on HRT-18G cells [33]. With this approach, we excluded five compounds from subsequent analysis because of cytotoxic effects on the HRT-18G cell line (Figure 4A). For the identification of potential IDV antagonists, HRT-18G cells were pretreated with the compounds for 1 h prior to infection with IDV^mNG^, during which 1 µM concentration for each compound was maintained. At 3 dpi, the number of mNG-positive cells was quantified to calculate the Z-score for each compound, and individual compounds with z-scores >1.5 or <−1.5 were considered hits (Figure 4B). Using this approach, we identified, next to our positive control Baloxavir Marboxil, a single compound inhibiting IDV replication, namely Apilimod, a PIKFyfe inhibitor [34]. Interestingly, we identified the JAK-inhibitor Ruxolitinib as a compound increasing IDV replication [35]. To determine the efficacy of the identified compounds, we treated HRT-18G cells with a serial dilution of the compounds of interest 1 h prior to adding the reporter virus. After 3 dpi, we counted the total number of mNG-positive cells under the different conditions and displayed the results using dose–response curves (Figure 4C,D). Baloxavir Marboxil and Apilimod displayed a dose-dependent effect with EC_50_ values of 0.36 µM and 0.06 µM, respectively, indicating that Apilimod is a more potent inhibitor of IDV replication than Baloxavir Marboxil. Intriguingly, Ruxolitinib showed increased mNG-positive cells at all concentrations, with an EC_50_ of 40 nM. This suggests that inhibiting the innate immune signaling can boost IDV^mNG^ replication on HRT-18G cells at relatively low concentrations. To avoid IDV^mNG^-specific artifacts, we repeated the dose–response experiment with the parental IDV^WT^ (Figure 4E). To detect virus-infected cells, we fixed the cells at the end of the incubation period and immunostained them for the NP protein. The results corroborated that Baloxavir Marboxil and Apilimod inhibit viral replication of IDVs, although the EC_50_ values were 3–4-fold higher (1.44 and 0.2 µM, respectively), suggesting that the reporter IDV is more susceptible to the inhibitors. Interestingly, we observed only a weak enhancing effect of Ruxolitinib on the viral replication of IDV^WT^, which suggests that this effect is specific to the reporter IDV.

Combined, these results underscore the versatility of our reporter IDV, serving as both a dependable means of monitoring viral replication progression and a valuable molecular tool for conducting inhibitor screening experiments.

## 4. Discussion

To the best of our knowledge, this is the first study reporting the generation of an Influenza D reporter virus that can be used to facilitate further molecular studies of IDV. After introducing the mNG fluorescent reporter gene into the NS segment of the IDV genome, we demonstrated that the resulting IDV^mNG^ and IDV^NS1-mNG^ reporter viruses could be rescued and reached high viral titers in HRT-18G cells. Serial passaging at different MOIs and monitoring mNG expression revealed that reporter expression could be detected in up to three consecutive passages. Finally, using the IDV^mNG^ reporter virus in a proof-of-concept screening, we identified Apilimod as a novel potent inhibitor of IDV replication, whereas the JAK-inhibitor Ruxolitinib was found to enhance IDV^mNG^ viral replication. Combined, we demonstrated the successful establishment of a new molecular tool for identifying novel antivirals against IDV, which can support further studies of the viral and host determinants involved in the broad host tropism of IDV.

In this study, we demonstrated that the NS1 segment of IDV can be utilized to introduce a reporter gene, similar to the strategy used for IAV and IBV [10,12,18], and can be applied to monitor viral replication within a single well over time. While we only created a reporter virus expressing the fluorescent mNG protein, this approach can likely also be used for incorporating a bioluminescence reporter with the NS1 segment. Such an approach would make it potentially possible to perform quantitative in vivo imaging in small animal models of IDV, such as mice and ferrets [1,11,36,37]. Furthermore, our results showed that the reporter virus with the C-terminal tagged NS1-mNG fusion protein generates infectious progeny virus, suggesting that this model can be applied to characterize the role of NS1 during the IDV viral replication cycle. In contrast, the N-terminal tagged mNG-NS2 recombinant virus did not lead to viable progeny virus, while the N-terminal peptide residue (10 AA) after post-translation autoproteolytic cleavage of the 2A motif of both the IDV^mNG^ and IDV^NS1-mNG^ did not seem to interfere. This suggests that the size of the N-terminal-fused protein can influence the function of NS2, such as its translocation from the cytoplasm to the nucleus, and/or binding to vRNPs and subsequent transfer out of the nucleus [17].

We demonstrated that the rescued reporter virus has comparable replication kinetics to the WT reverse genetics virus, although with some degree of attenuation as seen in the 1–2 orders of magnitude lower TCID_50_ values at all time points (Figure 1D). This phenotype is consistent with results obtained for similar reporter constructs for IAV [12]. One explanation for this phenotype might be the randomly accumulated deletions in both reporter constructs, as demonstrated by our full-segment sequencing approach. These random internal deletions in the reporter segment might lead to the generation of defective interfering (DI) particles that induce an innate immune response inhibiting viral replication, as previously described for IAV [38]. This might also explain why the JAK-inhibitor Ruxolitinib greatly enhanced viral replication for IDV^mNG^, but only minimally for IDV^WT^. As a possible solution to avoiding the accumulation of internal deletions in our reporter viruses, we performed plaque purification of the reporter virus to eliminate potentially low-abundant defective segments generated during viral rescue. In particular, we demonstrated that mNG gene expression is stably maintained in the genome in the early passages propagated at low MOI and that those viruses can be used to monitor the viral replication kinetics of IDV. We demonstrated that two consecutive rounds of plaque purification after the rescue limits the formation of DI particles, increasing the retention of the full-length fluorescent reporter segment.

By screening a panel of 80 compounds from the COVID box, we identified Apilimod as having a profound inhibitory effect on IDV replication, even more potent compared to our positive control Baloxavir Marboxil [31], which had an EC_50_ that was 6–7-fold higher. This compound is a known inhibitor of the lipid kinase PIKFyve, which is critical for maintaining the proper morphology of the cellular endosome/lysosome compartments. While Apilimod has initially been clinically evaluated to alleviate autoimmune diseases such as Crohn’s disease, it was also identified as a potential antiviral that blocks Ebola virus trafficking to its site of fusion and entry into the cytoplasm [34,39]. Whether Apilimod exhibits a similar antiviral mechanism in IDV and possibly other influenza viruses and retains this antiviral property in biologically relevant in vitro models of the respiratory epithelium of human, swine, and bovine remains to be evaluated [40].

Combined, these results demonstrate that the newly established recombinant IDV reporter virus can be applied to antiviral drug discovery and monitoring of viral replication, adding a new molecular tool for investigating IDV and paving the way for evaluating viral and host determinants involved in the broad host tropism.

## Figures and Tables

**Figure 1 viruses-15-02444-f001:**
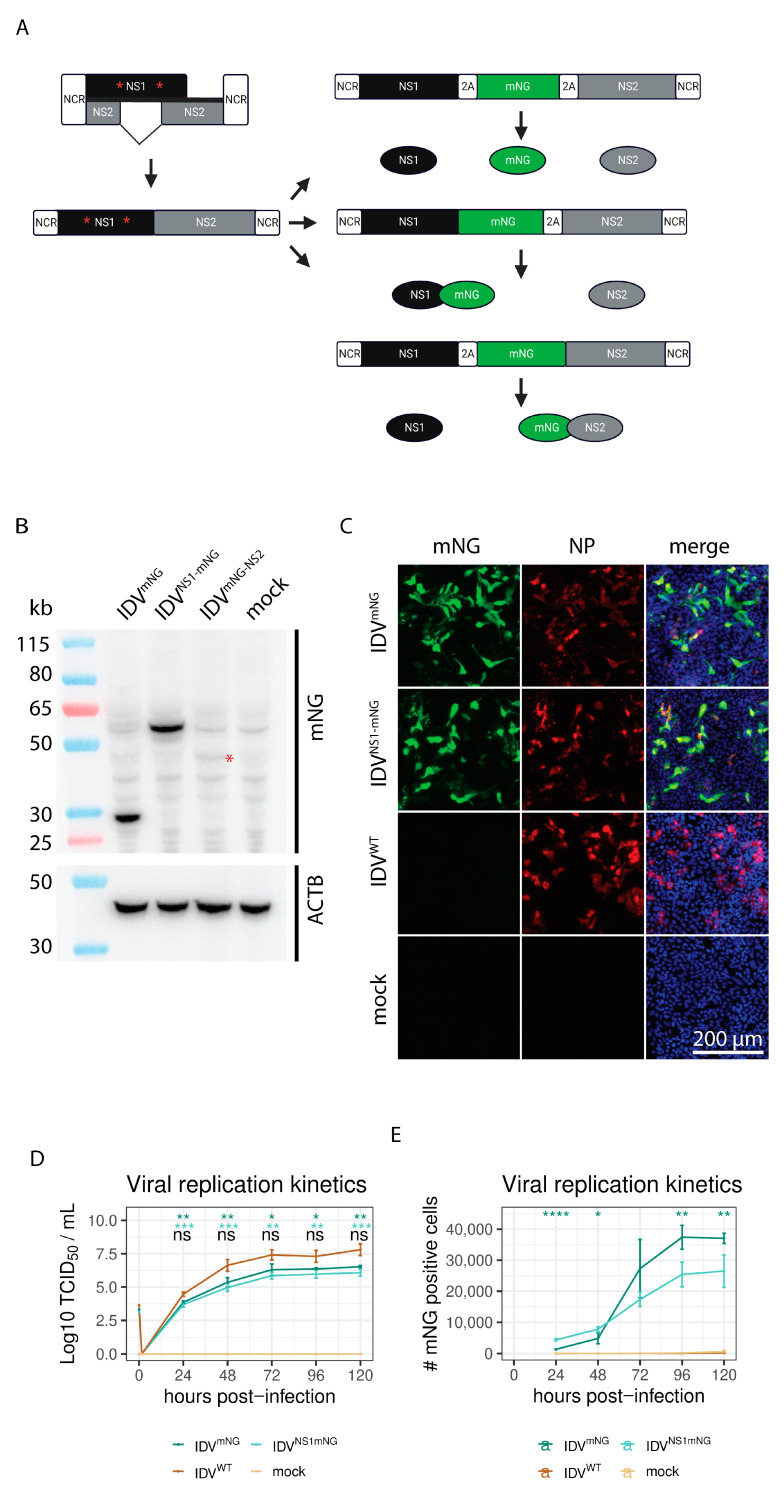
Rescue of recombinant influenza D reporter viruses. Schematic overview of the different reporter constructs generated, in which the mNeonGreen (mNG) gene (green) was inserted in between the coding sequences of NS1 (black) and NS2 (grey) and is flanked by one or two PTV-2A sequences (white). The synonymous point mutations to abrogate splicing of the NS1 gene are indicated with red asterixis. Created with BioRender.com (https://app.biorender.com/, accessed on 8 December 2023) (**A**). Western blot analysis using an mNG-directed antibody demonstrated the 2A autoproteolytic cleavage efficiency of the different reporter constructs. β-actin (ACTB) was used as a loading control. The red asterisk indicates the band corresponding to mNG-NS2 (**B**). Following virus rescue, HRT-18G cell cultures were infected with the different reporter viruses, using WT and mock-infected cultures as controls. At 72 hpi, cultures were formalin-fixed and processed for immunofluorescence analysis using an antibody against IDV (anti-NP, red) to visualize the overlap with mNG-expressing cells (green). The cell nuclei were counterstained with DAPI (blue). Scale bar is 200 µm (**C**). Corresponding viral replication kinetics were determined by monitoring progeny virus release every 24 h in the supernatant by virus titration (ns: not significant, *: *p* ≤ 0.05, **: *p* ≤ 0.01, ***: *p* ≤ 0.001, Tukey post-hoc test. Dark green: IDV^mNG^ vs. IDV^WT^, light green: IDV^NS1mNG^ vs. IDV^WT^, black: IDV^mNG^ vs. IDV^NS1mNG^) (**D**) and quantifying the mNG-positive HRT-18G cells (**E**) over time (*: *p* ≤ 0.05, **: *p* ≤ 0.01, ****: *p* ≤ 0.0001, Tukey post-hoc test, IDV^mNG^ vs. IDV^NS1mNG^).

**Figure 2 viruses-15-02444-f002:**
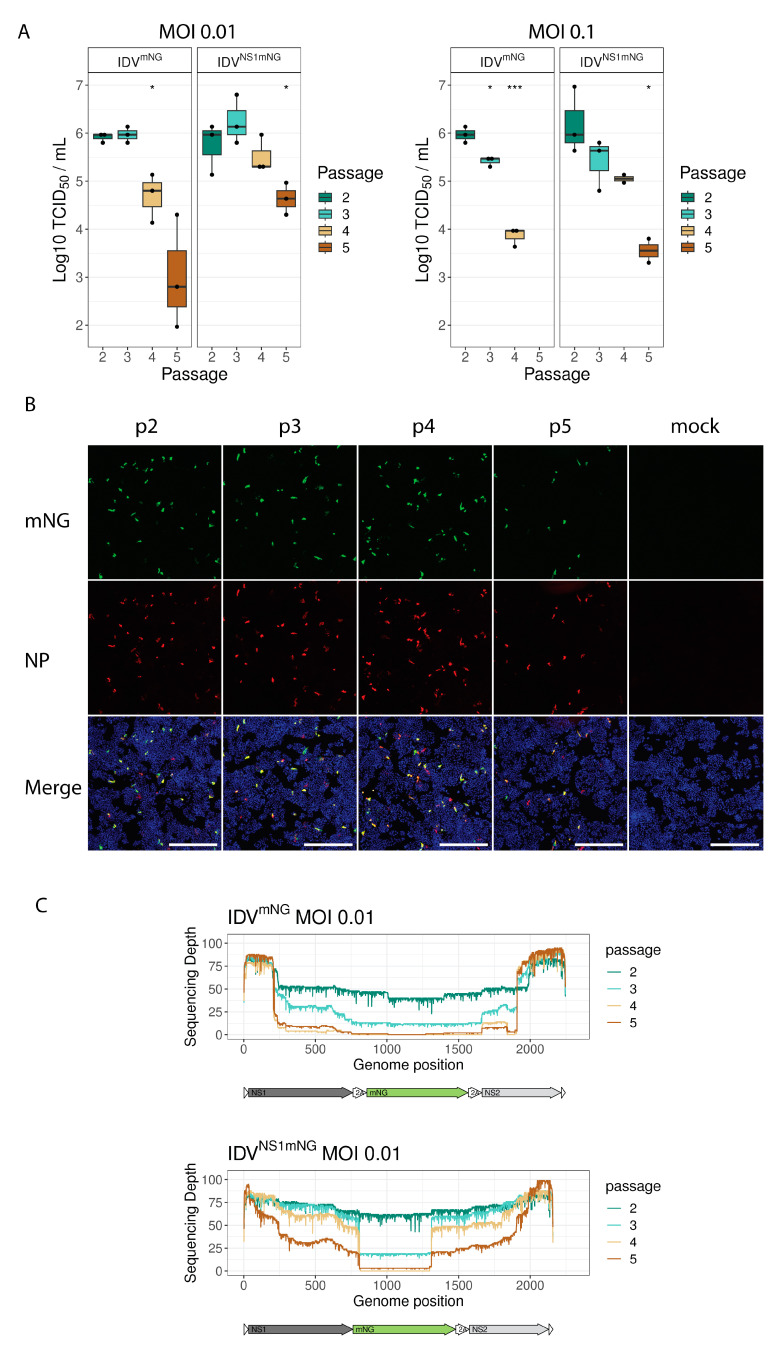
mNeonGreen expression is maintained for 3 passages in cell culture. Viral titers in the supernatant of HRT-18G cells infected with reporter IDVs after subsequent passages were determined at 144 hpi by assessing mNG fluorescence (*: *p* ≤ 0.05, ***: *p* ≤ 0.001, *t*-test against p2) (**A**). HRT-18G cells were infected with the passaged IDV^mNG^ and immunostained using an antibody against IDV (anti-NP, red). mNeonGreen-expressing cells are shown in green. Nuclei are stained with DAPI (blue). Scale bar is 500 µm (**B**). The NS segments of the passaged reporter IDVs were PCR-amplified and fully sequenced. The sequencing coverage per nucleotide (*y*-axis) of the NS segments of IDV (*x*-axis) after subsequent passages at MOI 0.01 is displayed in colored lines, showing an accumulation of deletions at later passages (**C**).

**Figure 3 viruses-15-02444-f003:**
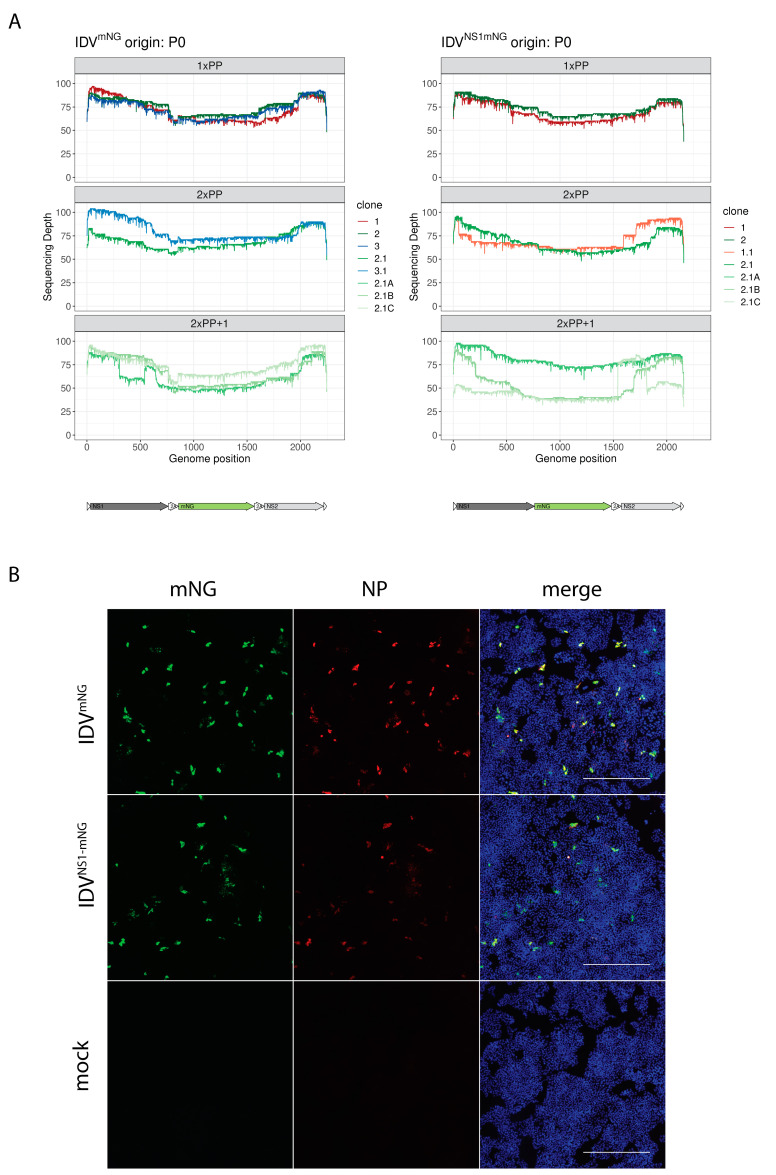
Plaque purification increases retention of the full-length reporter segment. The NS segment of the plaque-purified reporter IDVs was PCR amplified and fully sequenced. The sequencing coverage per nucleotide (*y*-axis) of the NS segments of IDV (*x*-axis) after subsequent plaque purification rounds and passages is displayed in colored lines, showing retention of the intact reporter segment over the passages. (1×PP: 1 round of plaque purification and subsequent expansion, 2×PP: 2 rounds of plaque purification and subsequent expansion, 2×PP+1: 2×PP with an additional passage at MOI 0.01). Viral descendants from plaque-purified clone 1 are depicted in shades of red, those descending from clone 2 in shades of green and from clone 3 in shades of blue. (**A**). HRT-18G cells were infected with the 2× plaque purified and passaged reporter viruses and immunostained using an antibody against IDV (anti-NP, red). mNeonGreen-expressing cells are shown in green. Nuclei are stained with DAPI (blue). Scale bar is 500 µm (**B**).

**Figure 4 viruses-15-02444-f004:**
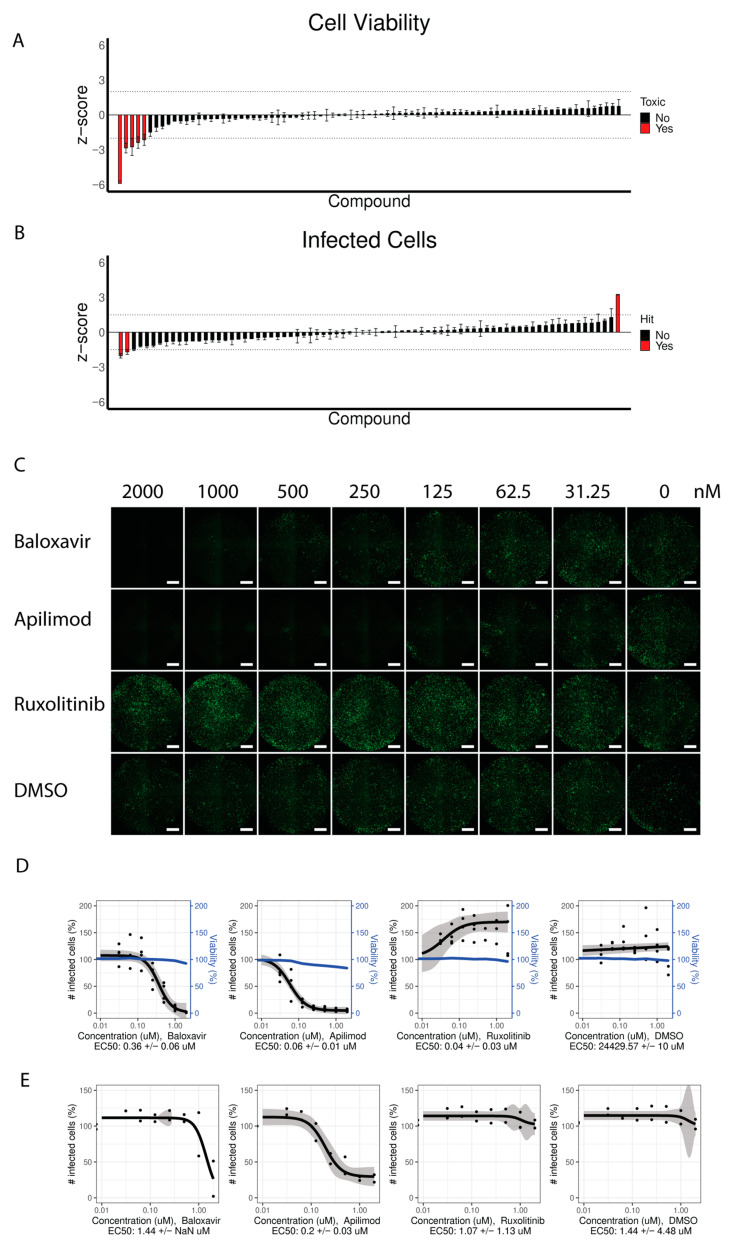
Use of the reporter IDV for inhibitor screens. A total of 80 compounds from the COVID box were assessed for their inhibitory effects against IDV replication. The tested panel of compounds was screened for viability after 73 h of treatment. Z-scores of cell viability were calculated for the plate (y-axis), and compounds with z-scores <−2 or >2 were considered toxic (red) and excluded from further analysis (**A**). Inhibitor screening was used to assess the efficacy of the panel of compounds against IDV. Z-scores for the number of infected mNG-positive cells at 72 hpi were calculated for the plate (y-axis), and compounds with z-scores <−1.5 or >1.5 were considered as hits (red) (**B**). The identified compounds were further assessed in a dose–response screen. Fluorescence images were generated for the cells treated with serial dilutions of the identified hit compounds and infected with reporter IDV^mNG^. Infected cells expressing mNG are shown in green. The scale bar is 1 mm (**C**). IDV^mNG^-infected cells were quantified and normalized to the untreated control (100%), and EC_50_ values were calculated (black, left y-axis, *n* = 4, points represent individual biological replicates, the shaded area is the 95% confidence interval). Cell viability was measured over the serial dilutions (blue, right y-axis) (**D**). Cells were treated with serial dilutions of the identified hit compounds and then infected with IDV^WT^. IDV^WT^-infected cells at 72 hpi were quantified and normalized to the untreated control (100%), and EC_50_ values were calculated (black, *n* = 2, points represent individual biological replicates, the shaded area is the 95% confidence interval) (**E**).

## Data Availability

The data presented in this study are available on request from the corresponding author.

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
