# Peer review of "Generation and Characterization of an Influenza D Reporter Virus"

_viruses, 2023, doi:10.3390/v15122444_

Round 1

Reviewer 1 Report

Comments and Suggestions for Authors

In the underlying manuscript Probst et al. describe – for the first time- the generation of an Influenza D reporter virus. Here, they introduced the green fluorescent protein mNeonGreen (mNG) into the IDV genome adapting a method, which had been established for Influenza A viruses. Finally, they show that their mNG-encoding IDV can be used to screen for antivirals.

Reporter viruses represent powerful tools in virology, especially for screening approaches e.g., to identify antivirals or host factors. Hence, this study represents a substantial advance for IDV research.

Major points

-       The mNG-encoding IDV is only genetically stable over 3 passages. As the authors point out in the discussion (line 427-433), plaque purification after virus rescue might yield a stable virus. As plaque purification is a standard procedure for the generation of recombinant IAV and IBV this reviewer thinks that the authors should plaque purify their mNG-encoding IDV and repeat the passaging experiment.

-       In Figure 1B the authors show that an mNG-NS2 fusions protein is not expressed upon transfection. In line with this, the correspondent reporter virus could not be rescued. This is surprising as IAV encoding fluorescent proteins fused to NS2/NEP could be successfully rescued. The authors should provide an explanation for the mNG-NS2 construct not be expressed upon transfection of a pol-II expression plasmid. Are cells transfected with this plasmid green fluorescent? 

-       Statistical analysis is missing throughout the paper and should be performed

Minor point

-       Superscripts in Figure 1B and C overlap with IDV

Author Response

Reviewer 1

We thank the reviewer for the thorough review of the article. We addressed all of their points:

The mNG-encoding IDV is only genetically stable over 3 passages. As the authors point out in the discussion (line 427-433), plaque purification after virus rescue might yield a stable virus. As plaque purification is a standard procedure for the generation of recombinant IAV and IBV this reviewer thinks that the authors should plaque purify their mNG-encoding IDV and repeat the passaging experiment.

Thank you for the suggestion, we have performed 2 rounds of plaque purification which showed an improvement in genomic stability. These results have been included in Figure 3.

In Figure 1B the authors show that an mNG-NS2 fusions protein is not expressed upon transfection. In line with this, the correspondent reporter virus could not be rescued. This is surprising as IAV encoding fluorescent proteins fused to NS2/NEP could be successfully rescued. The authors should provide an explanation for the mNG-NS2 construct not be expressed upon transfection of a pol-II expression plasmid. Are cells transfected with this plasmid green fluorescent?

The mNG-NS2 fusion protein was expressed upon transfection, but at lower levels than the NS1-mNG fusion protein or mNG alone (Figure 1B, highlighted with *), which later on did not yield in virus rescue or mNG positive cells (data not shown). As discussed in lines 462-467, we hypothesize that the fusion of mNG to the NS2 interrupts the function of NS2, and therefore, we could not rescue this virus.

Statistical analysis is missing throughout the paper and should be performed

Thank you for this comment, we have included the appropriate statistical analysis throughout the paper.

Superscripts in Figure 1B and C overlap with IDV

The text Fig 1b and c was corrected.

Reviewer 2 Report

Comments and Suggestions for Authors

The authors have successfully generated a reporter IDV incorporating a fluorescent protein by modifying the NS gene by reverse genetics. Although this reporter virus has low genetic stability upon passages, it could be used as a screening tool for candidate antiviral drugs. The authors need to address the points listed below before publication.

1. In all figures, the font sizes are too small to decipher. They need improvement.

2. Figure 2B and Figure 3C images are too dark to see. Change clear ones.

3. Both IDV-mNG and IDV-NS1mNG lacked large portion of NS coding regions after several passages. Discuss the reason about this and how to improve these defects.

4. Show strain name used in the Materials and Methods.

5. Line 42, The period was missing after “determinants”.

6. Line 310, “fluorescenceis” is a typographical error.

7. Line 343, "hRT-18G" is a typographical error.

8. Figure2B: Show data for NS1mNG virus.

9. Page numbers are missing in many references.

Comments on the Quality of English Language

Check typos.

Author Response

Reviewer 2

We thank the reviewer for the thorough review of the article. We addressed all of their points:

  1. In all figures, the font sizes are too small to decipher. They need improvement.
  2. Figure 2B and Figure 3C images are too dark to see. Change clear ones.

1,2: We improved the visibility of the figures, increased the brightness and increased the font sizes.

3. Both IDV-mNG and IDV-NS1mNG lacked large portion of NS coding regions after several passages. Discuss the reason about this and how to improve these defects.

3: During the revision, we did plaque purification to improve the stability of the virus. The new results are included in the results section and the updated discussion is found in lines 471-485.

4. Show strain name used in the Materials and Methods.

4: We added the strain name (D/CN286) in the Materials and Methods section.

5. Line 42, The period was missing after “determinants”.

6. Line 310, “fluorescenceis” is a typographical error.

7. Line 343, "hRT-18G" is a typographical error.

5,6,7: We corrected the typographical errors.

8. Figure2B: Show data for NS1mNG virus.

8: The data for NS1mNG was included in Sup Fig 1

9. Page numbers are missing in many references.

9: We corrected all the references.

Reviewer 3 Report

Comments and Suggestions for Authors

The manuscript entitled “Generation and characterization of an Influenza D reporter virus” is a well written study aiming at generating a fluorescent reporter IDV to facilitate further studies of IDV, such as antiviral screenings. The authors introduce the fluorescent reporter mNeonGreen (mNG) into the NS segment of IDV and demonstrate that the fluorescent gene is stably integrated for the first three passages after which deletions in the reporter gene accumulate. Furthermore, the authors use this reporter virus to rapidly screen a panel of compounds for antiviral properties and identify two compounds (Apilimod and Ruxolitinib) affecting IDV replication. I just have minor modifications suggestions:

1, In the abstract, the authors describe that “ These results demonstrate that the newly established recombinant IDV reporter virus can be applied for antiviral drug discovery and monitoring viral replication, …, and paving the way to evaluate determinants involved in the broad host tropism”, which is overstated. In this study, the authors have not conducted any experiments for investigation of the broad host tropism of IDV.

2. In the introduction, the citation “24” should be placed after “the fluorescent reporter mNeonGreen (mNG)”.

3, In the Materials and Methods, the authors describe that “The coding sequences for the reporter segments (NS1-2A-mNG-2A-NS2, NS1-mNG-2A-NS2, NS1-2A-mNG-NS2) were custom synthesized and cloned into a pcDNA3.1 backbone”. I am wondering if the authors include the non-coding regions of the IDV NS segment. Please clarify it.

4, In the Materials and Methods, the authors use the mNG fluorescent signal or IDV NP staining as readout to determine the viral TCID50 titers. Do the authors also use the hemagglutination activity as readout to determine the viral TCID50 titers? Please clarify it and include the corresponding data.

5, In the results, Figure 1A, what does the box between NS1 and NS2, 2A and NS2 represent? Please specify it in the picture.

6, In the results, Figure 1B, please indicate the size of all bands in the marker lane and modify the text on the top of the picture. In addition, the authors describe that there is a weak band corresponding to the mNG-NS2 fusion protein in Figure 1B. Please point out the band in the picture.

7, In the results, Figure 1C, please indicate the bar value and modify the text in the left of the picture; Figure 1E, it is better to show “% cells expressing mNG” rather than “mNG positive cells”.

8, In the Figures 3A and 3B, please reduce the thickness of axes, line/curve.

9, In the results, the IDVmNG of passage 2 with a MOI of 0.01 is used for antiviral compound screening. Do the authors use this MOI of IDVmNG virus to determine the EC50 of the identified compounds? How about the IDVwt used in the same dose-response experiment?

10, In the results, the authors state that, comparing to the IDVwt, the reporter IDV is more susceptible to the inhibitors. Since the titers of the IDVwt and IDVmNG may be determined by different readouts (the NP staining for the former one and the mNG fluorescence for the later one), the actual titers of the two viruses may be different, which is possible to result in the diverse susceptibility of the two viruses to the inhibitors. Please clarify it.

Author Response

Reviewer 3

We thank the reviewer for the thorough review of the article. We addressed all of their points:

1, In the abstract, the authors describe that “ These results demonstrate that the newly established recombinant IDV reporter virus can be applied for antiviral drug discovery and monitoring viral replication, …, and paving the way to evaluate determinants involved in the broad host tropism”, which is overstated. In this study, the authors have not conducted any experiments for investigation of the broad host tropism of IDV.

1: We removed the statement in the abstract.

2. In the introduction, the citation “24” should be placed after “the fluorescent reporter mNeonGreen (mNG)”.

2: We corrected the placement of the citation 24.

3, In the Materials and Methods, the authors describe that “The coding sequences for the reporter segments (NS1-2A-mNG-2A-NS2, NS1-mNG-2A-NS2, NS1-2A-mNG-NS2) were custom synthesized and cloned into a pcDNA3.1 backbone”. I am wondering if the authors include the non-coding regions of the IDV NS segment. Please clarify it.

3: We included the non-coding regions in the reporter segments since they are essential for the viral rescue. We updated the Materials and Methods section and Fig 1a to clarify.

4, In the Materials and Methods, the authors use the mNG fluorescent signal or IDV NP staining as readout to determine the viral TCID50 titers. Do the authors also use the hemagglutination activity as readout to determine the viral TCID50 titers? Please clarify it and include the corresponding data.

4: We did not use HA as a readout for the titration since the immunostaining is more sensitive and allows the simultaneous detection of both NP and the mNG reporter.

5, In the results, Figure 1A, what does the box between NS1 and NS2, 2A and NS2 represent? Please specify it in the picture.

5: In Figure 1a, the line inside the box representing NS2 represent the splice site. We removed it for better readability.

6, In the results, Figure 1B, please indicate the size of all bands in the marker lane and modify the text on the top of the picture. In addition, the authors describe that there is a weak band corresponding to the mNG-NS2 fusion protein in Figure 1B. Please point out the band in the picture.

6: We corrected the shifted band labels and included an arrow to indicate the band corresponding to mNG-NS2.

7, In the results, Figure 1C, please indicate the bar value and modify the text in the left of the picture; Figure 1E, it is better to show “% cells expressing mNG” rather than “mNG positive cells”.

7: The text has been improved, and the bar value has been added in Figure 1c. The kinetics in Figure 1e were determined in a live-cell imaging experiment, during which we, unfortunately, did not include any nuclear stain that would allow us to quantify the total number of cells in order to calculate the % of infected cells.

8, In the Figures 3A and 3B, please reduce the thickness of axes, line/curve.

8: The readability of the figure 3a and b were improved.

9, In the results, the IDVmNG of passage 2 with a MOI of 0.01 is used for antiviral compound screening. Do the authors use this MOI of IDVmNG virus to determine the EC50 of the identified compounds? How about the IDVwt used in the same dose-response experiment?

9: For the determination of the EC50, as well as for the infections with IDVWT, we used the same MOI to assure comparability. The Materials and Methods section was updated to clarify.

10, In the results, the authors state that, comparing to the IDVwt, the reporter IDV is more susceptible to the inhibitors. Since the titers of the IDVwt and IDVmNG may be determined by different readouts (the NP staining for the former one and the mNG fluorescence for the later one), the actual titers of the two viruses may be different, which is possible to result in the diverse susceptibility of the two viruses to the inhibitors. Please clarify it.

10: Thank you for the question; we hypothesize that this phenomenon is due to the lower replication kinetics for the reporter virus compared to IDVWT (shown in Figure 1D) and that this likely is additionally influenced by the accumulation of internal deletions, as mentioned in our discussion.